

# Achieving quantum advantage in a search for a violations of the Goldbach conjecture, with driven atoms in tailored potentials

Oleksandr V. Marchukov[1,2], Andrea Trombettoni[3,4],
Giuseppe Mussardo[4] and Maxim Olshanii[5⋆]

**1** Technische Universität Darmstadt, Institut für Angewandte Physik,
Hochschulstraße 4a, 64289 Darmstadt, Germany
**2** Institute of Photonics, Leibniz University Hannover,
Nienburger Straße 17, D-30167 Hannover, Germany
**3** Department of Physics, University of Trieste Strada Costiera 11, I-34151 Trieste, Italy
**4** SISSA and INFN, Sezione di Trieste, Via Bonomea 265, I-34136 Trieste, Italy
**5** Department of Physics, University of Massachusetts Boston,
Boston, Massachusetts 02125, USA

⋆ maxim.olchanyi@umb.edu

## Abstract

**The famous Goldbach conjecture states that any even natural number $N$ greater than 2 can be written as the sum of two prime numbers $p^{(I)}$ and $p^{(II)}$. In this article we propose a quantum analogue device that solves the following problem: *given a small prime $p^{(I)}$, identify a member $N$ of a $\mathcal{N}$-strong set even numbers for which $N - p^{(I)}$ is also a prime*. A table of suitable large primes $p^{(II)}$ is assumed to be known a priori. The device realizes the Grover quantum search protocol and as such ensures a $\sqrt{\mathcal{N}}$ quantum advantage. Our numerical example involves a set of 51 even numbers just above the highest even classical-numerically explored so far [T. O. e Silva, S. Herzog, and S. Pardi, Mathematics of Computation 83, 2033 (2013)]. For a given small prime number $p^{(I)} = 223$, it took our quantum algorithm 5 steps to identify the number $N = 4 \times 10^{18} + 14$ as featuring a Goldbach partition involving 223 and another prime, namely $p^{(II)} = 4 \times 10^{18} - 239$. Currently, our algorithm limits the number of evens to be tested simultaneously to $\mathcal{N} \sim \ln(N)$: larger samples will typically contain more than one even that can be partitioned with the help of a given $p^{(I)}$, thus leading to a departure from the Grover paradigm.**

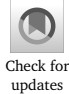

# 1  Introduction

## 1.1  Statement of the problem

The Goldbach conjecture [1] is a consequential [2–4] but yet unproven statement in number theory. It states that for every even natural number $N$, there exists a pair of prime numbers, $(p^{(I)}, p^{(II)})$ such that $N$ is a sum of these two numbers,

$$p^{(I)} + p^{(II)} = N.$$

We will use the convention that $p^{(I)} \leq p^{(II)}$ and call $(p^{(I)}, p^{(II)})$ a Goldbach pair. There can be more than one Goldbach pair for a given $N$.

   The Goldbach conjecture has been successfully verified numerically up to a maximal even number $N_0 = 4 \times 10^{18}$ [5]. Our goal is to offer a quantum speed-up to the future attempts to go beyond the current record.

## 1.2  The strategy

Imagine that one wants to test the even numbers that are even greater than $N_0$. One strategy would be to effectuate the following sieve:

   (i) Choose a large, $\mathcal{N}$-strong set of even numbers, $\{N_0 + 2n, n = 1, 2, \ldots \mathcal{N}\}$;

   (ii) Chose the lowest prime of interest, $p_2 = 3$ ($p_1 = 2$ can only Goldbach-partition $N = 4$) and exclude from the chosen even set all the evens for which $N - p_2$ is prime;

   (iii) Repeat step [(ii)] for the primes, $p_{m_I > 2}$, in an ascending order of $m_I$;

   (iv) Stop when no even numbers are left in the even set.

Observe that each $m$-labeled step in (iii) is an ideal setting for a Grover search.

## 1.3  Estimating the number of small primes to be used

One can address both the typical and the maximal number of the small primes to be used.

For typical number of primes, observe that a probability of $N - p_{m_{\mathrm{I}}}$ be prime can be estimated as

$$\mathrm{Prob}[N - p_{m_{\mathrm{I}}} \in \mathrm{primes}] \overset{N \gg p_m}{\sim} \mathrm{Prob}[N \in \mathrm{primes}] \sim \frac{d}{dN} \pi(N) \sim \frac{1}{\ln(N)},$$

where $\pi(N) \approx N/\ln(N)$ is the prime number function [1]. Hence for a typical $N$, it should be sufficient to try about $\ln(N)$ primes to find the one that contributes to one of its Goldbach pairs.

However, according to [6], some even numbers constitute rare exceptions, where their Goldbach primes are exceptionally large. It has been conjectured there that in the exceptional cases, the number of the primes to be tried is of the order of $\ln^2(N)$. These cases need to be treated separately.

### 1.4 Estimating the number of even numbers that can be tested simultaneously

Regretfully, in the current version of our algorithm, the number of evens, $\mathcal{N}$ is bounded from above. Indeed, let us fix the low prime $p^{(\mathrm{I})}$ to some $p_{m_{\mathrm{I}}}$ and start listing large even numbers $N$, one-by-one, in the ascending order, starting form large $N_0$. For a given $N$ and $p^{(\mathrm{I})}$, the probability that $N - p^{(\mathrm{I})}$ is a prime is $\sim 1/\ln(N)$. Hence for a sample larger than $\sim \ln(N)$, typically, there will be more than one even that can be partitioned with the help of a given $p^{(\mathrm{I})}$, thus leading to a departure from the Grover paradigm where there the database contains *only one* matching entry.

### 1.5 The state of the art

Our proposal assumes an ability to build a one-body potential with an arbitrarily tailored spectrum. By using holographic traps [7,8], such potentials recently became an experimental reality [9] with the implementation of the quantum prime potential for the first 15 prime numbers, i.e. of a potential $V(x)$ such $p^2/2m + V(x)$ has as eigenvalues the first 15 prime numbers (apart from an overall energy scale). In [9] the potential $v(x)$ giving the first 10 lucky primes.

The possibility of having a one-body potential opens the possibility of implementing schemes for translating number theory problems in quantum physical settings [10–21], including applications to few-body quantum prime factorization [11, 13–16, 18, 19]. Also in [21], we present a quantum proposal for testing the Goldbach conjecture, with no aspiration for a quantum advantage.

## 2 Protocol and its implementation

### 2.1 The Grover protocol

The canonical Grover database search protocol [22] allows, in approximately $\frac{\pi}{4}\sqrt{\mathcal{N}}$ steps, to identify a state $|\omega\rangle$ hidden in a unitary transformation

$$\hat{U}_\omega = \sum_{n'=1}^{\mathcal{N}} \left\{ \begin{array}{ll} -1, & \text{if } n' = \omega, \\ +1, & \text{otherwise} \end{array} \right\} |n'\rangle\langle n'|.$$

The protocol uses a state

$$|s\rangle \equiv \frac{1}{\sqrt{\mathcal{N}}} \sum_{n'=1}^{\mathcal{N}} |n'\rangle,$$

and the second unitary transformation

$$\hat{U}_s = \sum_{l=1}^{\mathcal{N}} \left\{ \begin{array}{ll} -1, & \text{if } l = s, \\ +1, & \text{otherwise} \end{array} \right\} |l\rangle\langle l|,$$

where $\{|l\rangle\}$ is an orthonormalized basis of which $|s\rangle$ is a member. Often, in literature, $\hat{U}_s$ has the opposite sign.

It can be shown that a two operator sequence $\hat{U}_\omega \hat{U}_s$, applied to the state $|s\rangle$

$$r(\mathcal{N}) \approx \frac{\pi}{4}\sqrt{\mathcal{N}}$$

times, leads to the state $|\omega\rangle$ sought after, with a probability close to unity:

$$\left(\hat{U}_s \hat{U}_\omega\right)^{r(\mathcal{N})} |s\rangle \approx (-1)^{r(\mathcal{N})} |\omega\rangle. \tag{1}$$

The sought key $|\omega\rangle$ can be read out using a state measurement in the $\{|n\rangle\}$ basis.

## 2.2  Hamiltonian

Our scheme involves a single atom with two internal states, $\| g\rangle$ and $\| e\rangle$, with two different potentials for each of the internal states. Following the Grover protocol, we alternate between two different additional pulsess: they will be realizing Grover operations $\hat{U}_\omega$ and $\hat{U}_s$ respectively.

The base Hamiltonian reads

$$\hat{H}_0 = U_0 \left\{ \sum_{n=n_{\min}}^{n_{\max}} N_n |n\rangle\langle n| \otimes \| g\rangle\langle g\| + \sum_{m_{\mathrm{II}}=(m_{\mathrm{II}})_{\min}}^{(m_{\mathrm{II}})_{\max}} p_{m_{\mathrm{II}}} |m_{\mathrm{II}}\rangle\langle m_{\mathrm{II}}| \otimes \| e\rangle\langle e\| \right\}, \tag{2}$$

with

$$\mathcal{N} \equiv n_{\max} - n_{\min} + 1,$$

being the size of the sample to be searched for violators of Goldbach conjecture. Here, $U_0$ is an overall energy scale, $N_n = 2n$ are the even numbers, and $p_m$ is a contiguous sequence of prime numbers greater than 2,

$$p_1 = 3; \quad p_2 = 5; \quad p_3 = 7; \quad p_4 = 11; \quad \dots$$

The two additional pulses are

$$\hat{V}^{(\omega)}(t) = 2V_0^{(\omega)} \cos(-p_{m_{\mathrm{I}}} U_0 t/\hbar) \left\{ \sum_{n=n_{\min}}^{n_{\max}} \sum_{m_{\mathrm{II}}=(m_{\mathrm{II}})_{\min}}^{(m_{\mathrm{II}})_{\max}} |m_{\mathrm{II}}\rangle\langle n| \otimes \| e\rangle\langle g\| + \text{h.c.} \right\}, \tag{3}$$

and

$$\hat{V}^{(s)} = V_0^{(s)} \sum_{n=n_{\min}}^{n_{\max}} \sum_{n'=n_{\min}}^{n_{\max}} |n'\rangle\langle n| \otimes \| g\rangle\langle g\|. \tag{4}$$

Notice that the first field, $\hat{V}^{(\omega)}(t)$, oscillates with a frequency proportional to the small prime of interest, while the second field, $\hat{V}^{(s)}$ does not evolve with time.

The operator $\hat{V}^{(\omega)}(t)$ couples every "ground state" $\| g\rangle$ (the "computational space") to every exited state $\| e\rangle$ ("auxiliary space"), with the same strength. The operator $\hat{V}^{(s)}$ couples every ground state with every other ground state, with the same strength.

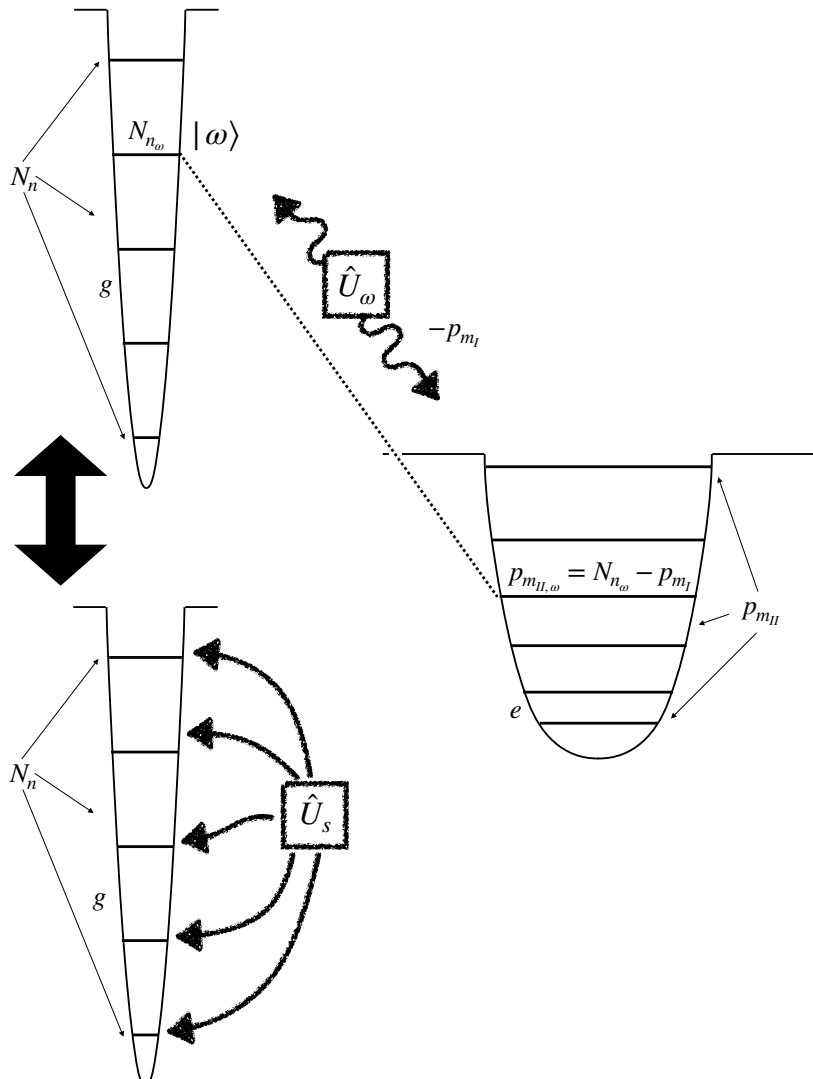

Figure 1: **The major ingredients of the protocol.** Energy is expressed in the units of $U_0$.

## 2.3 Initial state

The system is initialized in the state:

$$|s\rangle = \frac{1}{\sqrt{\mathcal{N}}} \sum_{n=n_{\min}}^{n_{\max}} |n\rangle \,.$$

This state is associated with the Grover state $|s\rangle$.

## 2.4 Realizing the $\omega$-gate

To realize the unitary $\hat{U}_\omega$ (see (3)), we chose the coupling $V_0^{(\omega)}$ to be exactly an even integer number of times smaller than the energy scale $U_0$:

$$V_0^{(\omega)} = \frac{U_0}{2M} \,. \tag{5}$$

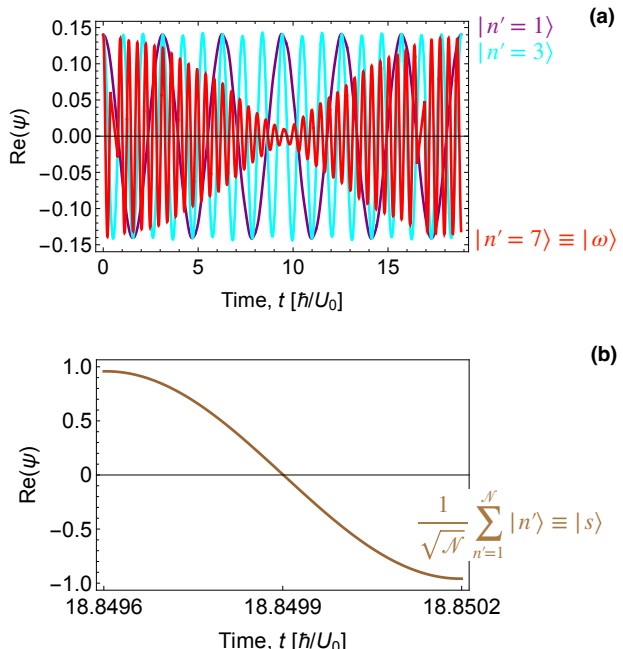

Figure 2: **Real part of the various components of the device wavefunction during the the first Grover cycle.** The computational space consists of $\mathcal{N} = 51$ states representing 51 even numbers just above $4 \times 10^{18}$, the so far largest number tested for violations of the Goldbach conjecture [5]. The auxiliary space consists all large prime candidates (7 total) for completion a Goldbach partition with any of the small primes $p_{m_{\mathrm{I}}}$ up to 307. The frequency of the pulse (3) corresponds to $p_{m_{\mathrm{I}}} = 233$. It identifies the 7'th member of the computational space, $N_{n_\omega} = 4 \times 10^{18} + 14$, as a number that satisfies the Goldbach conjecture, with the second prime being $p_{m_{\mathrm{II},\omega}} = 4 \times 10^{18} - 209$. (a) Our realization of the Grover $\omega$-pulse. For the magnitude of the $2\pi$-pulse, we chose $V_0^{(\omega)} = \frac{1}{6} U_0$. As by design, the sign in front of the Grover matching entry $|\omega\rangle$ changes by the end of the pulse, while the amplitudes of the remaining members of the computational space remain the same. (b) Our realization of the Grover $s$-pulse. We choose $V_0^{(s)} = 100. U_0$. Observe that indeed, the prefactor in from of the Grover $|s\rangle$-state changes while its absolute value does not. Here, $n' \equiv n - n_{\min} + 1$.

We will also assume that the pulse is weak:

$$M \gg 1.$$

To realize the Grover gate $\hat{U}_\omega$, the pulse (3) will be applied for a time

$$\tau^{(\omega)} = \pi \frac{\hbar}{V_0^{(\omega)}}.$$

Recall that the prefactor in front of this pulse oscillates with a frequency proportional to a small prime $p_{m_{\mathrm{I}}}$. If the excited state spectrum contains a large prime $p_{m_{\mathrm{II},\omega}}$ that completes a Goldbach pair,

$$N_{n_\omega} = p_{m_{\mathrm{I}}} + p_{m_{\mathrm{II},\omega}},$$

for some $m_{\mathrm{II},\omega}$, the pulse $\hat{V}^{(\omega)}(t)$ will effectuate a $2\pi$-pulse involving the states $|n_\omega\rangle \otimes \| g\rangle$ and $|m_{\mathrm{II},\omega}\rangle \otimes \| e\rangle$; that is, by the end of the pulse, the prefactor in front of the matching state $|n_\omega\rangle \otimes \| g\rangle$ will change sign. The rest of the $n$-states will not be significantly perturbed thus

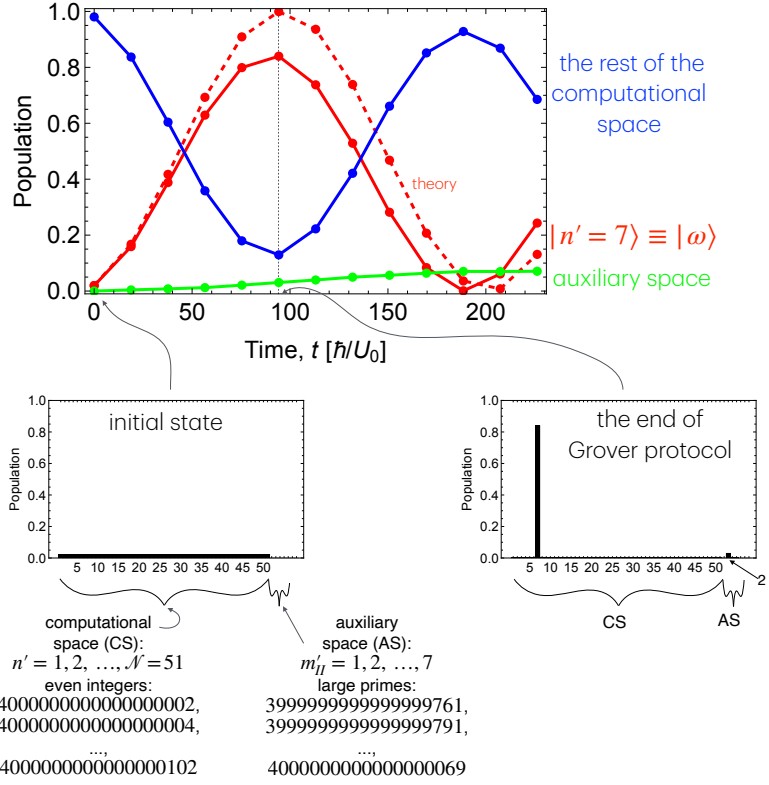

Figure 3: **Grover protocol**. Results of a computer simulation of the Grover sequence. System structure and the input parameters are the same as at Fig. 2. The Grover target state, $|n' = 7\rangle \equiv |\omega\rangle$ is the even number $N_{n_\omega} = 4 \times 10^{18} + 14$ that is being identified as a sum of two primes, $p_{m_I} = 233$ and $p_{m_{II,\omega}} = 4 \times 10^{18} - 209$. Red dashed line with symbols is the theoretical prediction for the population of the matching entry bin, $|\langle \omega | \psi \rangle|^2 = \cos^2((2n + 1)\delta\theta)$, where $n$ is the number of Grover iterations, $\delta\Theta = \arcsin(1/\sqrt{\mathcal{N}})$, and $\mathcal{N} = 51$ is the size of the computational space. It is evident that $n = 5$ is the theoretically predicted optimal number of iterations. Indeed, numerically the matching entry state will be detected in a state measurement with a probability of about 80%. Notice that unlike in the idealized scenario, the $m'_{II,\omega} = 2$ auxiliary state, corresponding to $p_{m_{II,\omega}}$, remains weakly populated. We also propagate the procedure for seven more steps to further compare the numerical results with theoretical predictions. Solid red line with symbols: population of the matching entry state in the end of every Grover iteration, plus it's initial value. Solid blue line with symbols: same, but for the rest of the computational space. Solid green line with symbols: same, but for the auxiliary space. Here, $n' \equiv n - n_{\min} + 1$ and $m'_{II} \equiv m_{II} - (m_{II})_{\min} + 1$.

propagating freely, for a period of time that is commensurate (thanks to the condition (5)) with the shortest free oscillation period, $\hbar/U_0$. Hence, by the end the pulse, two things will happen: (a) the prefactor in front of $|n_\omega\rangle$ will change sign; (b) the prefactor in front of all the other $|n\rangle$ states will return to the respective values they had in the beginning of the pulse.[1] These transformations indeed constitute a $\omega$-gate.

---

[1] A similar prefactor will also exist for the $|n_\omega\rangle$ state.

## 2.5 Realizing the $s$-gate

The operator $\hat{V}^{(s)}$ (see (4)) is proportional to a projector to the $|s\rangle$ state:

$$\hat{V}^{(s)} = \mathcal{N} V_0^{(s)} |s\rangle\langle s| \,.$$

To realize the unitary $\hat{U}_s$, we use a large value of the coupling constant and apply $\hat{V}^{(s)}$ for a short time:

$$\mathcal{N} V_0^{(s)} \gg U_0 \,,$$
$$\tau^{(s)} = \pi \frac{\hbar}{\mathcal{N} V_0^{(s)}} \ll \frac{\hbar}{U_0} \,.$$

The base Hamiltonian (2) will remain present (in particular, to keep atoms trapped), but its contribution to the dynamics will be negligible.

## 2.6 Readout

In the end, we measure the atom state in the $\{|n\rangle\} \otimes \| g\rangle$ basis. If a Goldbach protocol is realized successfully, the detected state $|n_\omega\rangle$ will indicate that the corresponding even number $N_{n_\omega}$ has a Goldbach partition that features $p_{m_\omega}$.

## 3 Numerical results

Our computational space covers 51 even numbers just above the largest number tested for violations of the Goldbach conjecture numerically [5]: they cover a range from $4 \times 10^{18} + 2$ through $4 \times 10^{18} + 102$. From this range, there are 7 large primes that can in principle contribute to a Goldbach partition of any of the members of the computational space, if small primes from 3 through 307 are used. These large primes lie from $4 \times 10^{18} - 309$ through $4 \times 10^{18} + 60$. The small prime that has been used was 223.

It is important to realize that both the $\omega$- and $s$ pulse realizations that we propose, effectuate these pulses only approximately. Recall that in an ideal $\omega$-gate, the field $\hat{V}^{(\omega)}$ would only have the transition matrix elements between $|n_\omega\rangle \otimes \| g\rangle$ and $|m_{\mathrm{II},\omega}\rangle \times \| e\rangle$. However, in our realization, each $|n\rangle \otimes \| g\rangle$ state is coupled to every $|m_{\mathrm{II}}\rangle \times \| e\rangle$ state, albeit in an off-resonant manner. Likewise, during the operation of the $s$-gate, one needs to keep atoms trapped by the base field $\hat{H}_0$. It's presence will effectively modify the matrix elements of the $\hat{U}_s$ operator. A portion of this project was devoted to the optimization of the pulse parameters aimed at a mitigation of the unwanted effects mentioned above.

We found the parameter choice

$$V_0^{(\omega)} = \frac{1}{6} U_0 \,,$$
$$V_0^{(s)} = 100. \, U_0 \,,$$

to be optimal for our system. Figs. 2(a) and (b) illustrate the result of this optimization. The goal of the $\hat{V}_\omega$ pulse is to flip the sign of the coefficient in front of the Grover's database key; in our case, the latter is represented by the state $|n_\omega\rangle \otimes \| g\rangle$. Indeed, the red line at Fig. (2)(a), representing the real part of the coefficient in front of $|n_\omega\rangle \otimes \| g\rangle$ switches form its initial value of 0.14 to its opposite, $-0.14$. The rest of the basis stes return to their initial values by the end of the pulse. Fig. (2)(b) performs a similar analysis for the $|s\rangle$ state whose prefactor is also supposed to change sign by the end of the $\hat{V}_s$ pulse.

Finally, Fig. 3 describes the entirety of the Grover protocol described in Section 2. That Figure traces the population of the "database key" state, along with the total population of the remaining computational space states and the total population of the auxiliary manifold. Theoretically, five Grover cycles constitute an ideal sequence in our case (see Fig. 3). At this moment, a Goldbach decomposition

$$400000000000000014 = 223 + 3999999999999999761 \,,$$

will be detected with a probability of about 80%. As we mentioned in the beginning of this section, one of the probable reasons for the 20% loss of accuracy is the off-resonant transitions during the $\hat{V}_\omega$-pulses and the parasitic presence of the base Hamiltonian $\hat{H}_0$ during the operation of the $s$-gate.

## 4 Conclusion and outlook

In this article, we suggested an analogue device that allows for a quantum polynomial speed-up of a search for a large even number $N_{n_\omega}$ such that a given small prime $p^{(I)}$ contributes to one of its Goldbach partitions. The number $N_{n_\omega}$ is known to belong to a set $\{N_0+2n, n = 1, 2, \ldots \mathcal{N}\}$. A table of suitable large primes $p^{(II)}$ is assumed to be known a priori. Our device realizes the Grover search algorithm verbatim: not surprisingly, it offers an $\sqrt{\mathcal{N}}$ speed up as compared to a brute force classical search.

In our numerical example, our device was able to identify, in five steps, from a set of 51 evens just above the highest even classical-numerically explored so far [5], the number $4 \times 10^{18} + 14$ as being Goldbach partitionable using a given small prime $p^{(I)} = 223$ and an a priori unknown large prime.

This work was directly inspired by the recent advances in generating atomic potentials with predetermined spectra [9]. While in our proposal, we use atoms with a two-level internal structure, in practical applications both ground and excited spectra can be combines in a spectrum of a single potential, provided that the two groups are energetically well-separated.

Our scheme assumes a high degree of a control over the values of the matrix elements of the $\hat{V}^{(\omega)}$ and $\hat{V}^{(s)}$. Gaining such control constitutes one of the future directions of our research; some preliminary steps can be found in [20]. An additional challenge is to find a way to realize a singular pulse used at the Grover diffusion step. Singular additions to the two-dimensional billiards has been studied in the context of quantum chaos [23–25]. For one-dimensional systems, one example is a $\delta$-functional barrier in the center of an infinitely deep square well, in the subspace spanned by the even eigenstates.

Another direction of future research is to explore a possibility of working with sets of evens that can contain more than one number partitionable by a given small prime. The theory of Grover searches of sets containing more than one matching object is well developed [26]. Note that contrary to the standard problem of finding *a* matching object, we need to find *all* of them.

## Acknowledgments

A significant portion of this work was produced during the thematic trimester on "Quantum Many-Body Systems Out-of-Equilibrium", at the Institut Henri Poincaré (Paris): AT, GM, and MO are wholeheartedly grateful to the organizers of the trimester, Rosario Fazio, Thierry Giamarchi, Anna Minguzzi, and Patrizia Vignolo, for the opportunity to be part of the program.

**Funding information** OVM acknowledges the support by the DLR German Aerospace Center with funds provided by the Federal Ministry for Economic Affairs and Energy (BMWi) under Grant No. 50WM1957 and No. 50WM2250E. MO was supported by the NSF Grant No. PHY-2309271. The authors would like to thank the Institut Henri Poincaré (UAR 839 CNRS-Sorbonne Université) and the LabEx CARMIN (ANR-10-LABX-59-01) for their support. - and the Institut Henri Poincaré (UAR 839 CNRS-Sorbonne Université) and LabEx CARMIN (ANR-10-LABX-59-01).

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
