# Peer review of "Achieving quantum advantage in a search for a violations of the Goldbach conjecture, with driven atoms in tailored potentials"

_SciPost Physics Core, doi:SciPost Phys. Core 8, 074 (2025)_

## Round 2 · Referee Report · Anonymous (Referee 1) · 2024-10-23

Strengths

1- Relevance. In their work, the authors investigate the possibility of using a modified Grover algorithm to test Goldbach's conjecture, which states that every natural number greater than 2 can be written as the sum of two prime numbers. The authors' idea, which seems well-based, is that, thanks to the use of a quantum algorithm, the search process would be extremely faster. This, given the ever-increasing relevance that prime numbers have in various fields of modern science, should justify the publication of their paper.

Weaknesses

1- From a scientific point of view, I am puzzled by the fact that the authors provide no evidence, neither analytical nor numerical, that their modified version of Grover's algorithm continues to correctly converge to the searched state with the same rate. Perhaps the authors consider it sufficient that the algorithm is similar to a well-known one to ensure that everything goes smoothly. However, the change in the algorithm is not trivial and doubts about the convergence may arise. The authors should make an effort to make their claims more solid.

2- Even less do the authors take the trouble to support with analytical and/or numerical evidence the fact that the dynamics of the systems they considered (equations 6 and 7) correctly simulates their algorithm. This is not a trivial point since the authors make several assumptions, and it would be interesting to test them numerically. To provide an example, the authors require that some quantities must be much smaller than 1 (non enumerated equations). Great! Is 10^-1 enough? Or do we need to reach 10^-10? And how are these values ​​affected by the other parameters of the system? These seem to me to be interesting questions that are not answered within their paper.

3- The paper seems to have been rushed out to publication. This feeling comes from several small problems. For example, on page 2, the authors quickly summarize the Grover algorithm in its classic version. After equation 4 they write: "The sought key |w> be read out using the state measurement in the {|n>} \otimes |g > basis". What is the state |g >? It has never been defined before. Moreover, what would be its role in a Grover algorithm? After a few pages, we see that the state |g > is nothing more than one of the two internal states of a two-level system introduced to realize a system whose dynamics should simulate the modified Grover algorithm that they introduce. OK, but why insert it pages before in a cryptic sentence that only adds confusion?

4- Of course, if this were the only problem, it would be a small oversight that happens to everyone. The paper presents a huge amount of typos in various mathematical expressions, especially those in the text. They range from "|m_{max}>>" with the double >> to an artistic "|n′<\otimes|e>", etc. Before resubmitting the paper, the authors should do a thorough job of cleaning up these errors.

5- I would also suggest extending the introduction of the paper to better frame the problem and make it accessible to a wider audience of readers.

Report

Even if the topic is relevant and the results seems to be of wide interest, I cannot recommend publishing the paper in this version. However, once that the problems of the paper will be fixed, after a thorough rewriting, an improved version of their work will deserve a publication.

Requested changes

1- Provide a proof that the modified algorithm that they introduced, converges at the right state with a rate comparable with the original Grover's one.

2- Provide a test of the stability of the dynamics in the system introduced in equations 6 and 7 for different choices of parameters

3- Subject the text of the article to a careful rewriting that better clarifies the various points and removes the numerous typos

4- Extend the introduction to make the paper more accessible to a wider audience

Recommendation

Ask for major revision

---

## Round 3 · Referee Report · Anonymous (Referee 2) · 2025-9-7

Strengths
1- Insightful implementation of Grover quantum search 2- Sound analysis of the efficiency
Weaknesses
1- Numerical results are not clearly explained
Report
In my view the authors addressed the concerns of the previous referee. The paper can be accepted for publication after the authors have considered the following points:
-
The potential in Eq. (4) is supposed to implement an all-to-all coupling between the states of the databased. However, it does not seem to realise the diffusion operator of Grover search $\hat U_s$. In fact, in the model of Eq. (2) the states of the database have different energies. Thus, since (4) is a perturbation, the effective coupling between a given pair of states will also depend on the energy difference. The authors shall comment how this affects the final efficiency of the protocol.
-
The numerical results are not clearly explained. It is difficult to extract what the figures show (which quantity is shown? What does it tell about the efficiency of the protocol? etc)- a text in Sec. 3 that guides the reader through the figures and discusses the findings would significantly increase the clarity of the presentation.
-
Fig.3: What limits the efficiency of the protocol to 80%?
Requested changes
In addition to addressing the points listed above, I recommend that the authors fix several typos (e.g., "exited"->"excited", "t hey" ->"they", "targrert"->"target", "hight"->"high").
Recommendation
Ask for minor revision

Author: Maxim Olshanii on 2025-09-19 [id 5837]
(in reply to Report 1 on 2025-09-07)To start let us mention that without the referees, this paper would be two time shorter. Their comments were relevant, focused, and inspiring.
We added several sentences of a comment, in the end of the Section 2.5. Additionally, we completely eliminated the misleading notion of "perturbation."
We expanded the Chapter 3 to better describe our numerical results.
We added two small paragraphs to the ens of the Chapter 3.

---

## Round 3 · Author Response

Our referee remarks inspired us to, effectively, rewrite the paper and run a series of computer simulations, for which we are indefinitely grateful.
-
We modified the protocol in such a way that it is now isomorphic to the conventional Grover search scheme. Extensive literature exists that studies the efficiency of the latter.
-
We underwent an extensive numerical experimentation cycle and found an optimal set of system parameters
2b. On the negative side, our numerical study showed that the requirements to the relative values of the perturbation matrix elements, for both omega- and s-gates, are much more stringent than we expected. While the unperturbed Hamiltonian remains firmly rooted in potentials realized in Donatella Cassettari's lab, the gate perturbations used in our numerics are represented by their idealized versions. More work is needed to bring our proposal in contact with the AMO reality. Our recent AVS Quantum [O. V. Marchukov and M. Olshanii, AVS Quantum Science 7, 013801 (2025)]is a step in this direction.
-
The text is completely rewritten.
-
Our introduction is substantially extended.

---

## Round 3 · List of Changes

Major changes
- We added numerical simulations.
- The protocol is mildly altered: in the most recent version, even numbers and large primes correspond to the quantum eigenstates and small primes are perturbation frequencies. As the result, the protocol is now identical to the Grover one.
- Overall, the text has been very substantially rewritten.

---

## Round 4 · Author Response

To start let us mention that without the referees, this paper would be two time shorter. Their comments were relevant, focused, and inspiring.

---

## Round 4 · List of Changes

We added several sentences of a comment, in the end of the Section 2.5. Additionally, we completely eliminated the misleading notion of "perturbation."

We expanded the Chapter 3 to better describe our numerical results.

We added two small paragraphs to the ens of the Chapter 3.

---

## Round 5 · Author Response

Many thanks to the Editorial Board and the Referees. Our manuscript is now unrecognizably better.

---

## Round 5 · List of Changes

• Misprints corrected
  • Section 3 is fully restructured

---

## Editorial Decision

published